# Evaluating the structure-based virtual screening performance of SARS-CoV-2 main protease: A benchmarking approach and a multistage screening example against the wild-type and Omicron variants

**Noha Galal[1,2], Botros Y. Beshay[3], Omar Soliman[4], Muhammad I. Ismail[5], Mohamed Abdelfadil[6], Mohamed El-Hadidi[7,8], Reem K. Arafa[1,2]*, Tamer M. Ibrahim[6,8]***

**1** Drug Design and Discovery Laboratory, Zewail City of Science and Technology, October Gardens, 6th of October City, Giza, Egypt, **2** Biomedical Sciences Program, UST, Zewail City of Science and Technology, October Gardens, 6th of October City, Giza, Egypt, **3** Pharmaceutical Chemistry Department (Pharmaceutical Sciences Division), College of Pharmacy, Arab Academy for Science, Technology and Maritime Transport, Alexandria, Egypt, **4** Genomics Program, Children's Cancer Hospital Egypt, Cairo, Egypt, **5** Department of Pharmaceutical Chemistry, Faculty of Pharmacy, The British University in Egypt, El-Sherouk City, Egypt, **6** Department of Pharmaceutical Chemistry, Faculty of Pharmacy, Kafrelsheikh University, Kafrelsheikh, Egypt, **7** Institute of Cancer and Genomic Sciences (ICGS), School of Medical Sciences, College of Medicine and Health, University of Birmingham Dubai, Dubai, United Arab Emirates, **8** Center for Informatics Science (CIS), School of Information Technology and Computer Science (ITCS), Nile University, Giza, Egypt

☯ These authors contributed equally to this work
* rkhidr@zewailcity.edu.eg (RKA); tamer_mohamad@pharm.kfs.edu.eg (TMI)

## Abstract

COVID-19 still poses a worldwide health threat due to continuous viral mutations and potential resistance to vaccination. SARS-CoV-2 viral multiplication hindrance by inhibiting the viral main protease (Mpro) deemed propitious. Structure-based virtual screening (SBVS) is a conventional strategy for discovering new inhibitors. Nonetheless, the SBVS efforts against Mpro variants needed to be benchmarked. Herein, in the first stage of the study, we evaluated four docking tools (FRED, PLANTS, AutoDock Vina and CDOCKER) *via* an in-depth benchmarking approach against SARS-CoV2 Mpro of both the wild type (WTMpro) and the deadly Omicron P132H variant (OMpro). We started by compiling an active dataset of non-covalent small molecule inhibitors of the WTMpro from literature and the COVID-Moonshot database along with generating a high-quality benchmark set *via* DEKOIS 2.0. pROC-Chemotype plots revealed superior performance for AutoDock Vina against WTMpro, while both FRED and AutoDock Vina demonstrated excellent performance for OMPro. In the second stage, VS was performed on a focused library of 636 compounds transformed from the early-enriched cluster related to perampanel via a scaffold hopping approach. Subsequently, molecular dynamics (MD) simulation and MM GBSA calculations validated the binding potential of the recommended hits against both explored targets. This study provides an example of how to conduct an in-depth benchmarking approach for both WTMPro and OMPro variants and offering an evaluated SBVS protocol for them both.

**Data availability statement:** "The SMILES of the active and decoy molecules used in docking benchmarking are within the Supporting Information files. Codes for docking bench-marking are available as follows: AudoDock Vina docking and file preparation [https://vina.scripps.edu/manual/; https://ccsb.scripps.edu/mgltools/documentation/], PLANTS docking [https://github.com/purnawanpp/plants?tab=readme-ov-file], and FRED docking and preparation [https://docs.eyesopen.com/applications/oedocking/fred/fred.html]. Please note, the CDOCKER module (for CDOCKER docking) is property of "Dassault Systèmes BIOVIA," therefore access to it might be restricted. Codes for molecular dynamics GROMACS and gmx MMPBSA codes are available at https://manual.gromacs.org/2023.3/manual-2023.3.pdf, https://github.com/Valdes-Tresanco-MS/gmx_MMPBSA. The transformed focused library was downloaded from https://drugspacex.simm.ac.cn/help/ [DOI: https://doi.org/10.1093/nar/gkaa920]. All other relevant data are within the paper and its Supporting information files."

**Funding:** The author(s) received no specific funding for this work.

**Competing interests:** The authors have declared that no competing interests exist.

## Introduction

With the advent of the emergence of the COVID-19 pandemic mediated by the SARS-CoV-2 virus, the scientific arena found itself facing an unprecedented global health threat posing enigmatic challenges and pending swift resolution to attend to the daunting figures of disease spread and mortalities[1]. This was further aggravated by the inherent rapid mutations of the virus leading to new variants, such as. Alpha (B.1.1.7) [2,3], Beta (B.1.351) [4,5], Gamma (P.1) [6], and Delta (B.1.617.2) [7,8]; and particularly Omicron (B.1.1.529) [9,10]. Importantly, the Omicron variant is associated with higher risk of transmission, potential drug resistance mutations, and possible resistance to vaccination. As previously identified for other coronaviruses, SARS-CoV-2 contains a set of structural proteins (SPs) that are important for producing a complete viral particle, additional non-structural proteins (NSPs) that act as enzymes or transcription/replication factors in the viral life cycle, and other numerous accessory proteins. One essential NSP for viral replication is the SARS-CoV-2 main protease (Mpro) (also known as 3Clpro, or NSP5) whose proteolytic cleavage of polyproteins generates functional polypeptides [11,12]. Yet, on the basis of consideration to the emergence of drug resistance to drug-based interventions, the scientific community is vigilantly monitoring antivirals' potential mechanisms of drug resistance against Mpro, particularly that observed by the Omicron variant, such as the P132H mutant variant [13,14].

For intervening with the function of Mpro, the non-covalent, nonpeptidic inhibitors option is preferred in many drug discovery campaigns to avoid potential serious side effects arising from off-target irreversible interactions and altered pharmacokinetic profile [15]. Accordingly, extensive research employing virtual screening and design approaches revealed potent and non-peptidic inhibitors with optimized PK profiles such as ensitrelvir (S-217622) [16,17].

In the context of structure-based virtual screening (SBVS), benchmarking is an evaluation approach of the screening performance of docking tools. Benchmark datasets consist of bioactive molecules and structurally similar inactive decoys for a specific protein target [18,19]. The effectiveness of a docking tool in SBVS is measured by its capacity to prioritize known bioactive molecules over decoys. Therefore, benchmarking is a valuable method for identifying suitable virtual screening pipelines with improved predictive capabilities [20–22].

### Rational of the study

The aim of the current study is to provide an in-depth benchmarking investigation for popular docking tools on both WTMpro and the mutant OMPro (P132H) variant, and accordingly, offering an example of a prospective virtual screening approach on both Mpro variants, as illustrated in **Fig 1**. We started by compiling an active data set of noncovalent, non-peptidomimetic inhibitors, emphasizing the diversity and potency of small inhibitors of the Mpro reported from both literature and the COVID-Moonshot to produce high quality DEKOIS 2.0 benchmark set. Subsequently, we probed the benchmarking performance of four docking tools employing structures of Mpro of both the wild-type (WTMpro) and the Omicron P132H variant (OMpro) to stand on the most appropriate tool for VS against both targets. Grounded on the benchmarking results and the chemotype analysis, we subsequently built a virtual new focused chemical library featuring the scaffold of the top cluster (perampanel, pyridone derivatives in this study) to propose new plausible Mpro inhibitors. Thereafter, the recommended hits from the VS campaign were further validated *via* MD simulations and MM GBSA analysis.

## Methods

### Active set selection and generation of DEKOIS 2.0 benchmark set

Filtration and clustering of our active set were performed using Data-Warrior software [23] resulting in 19 clusters adjusting fragment fingerprint as a descriptor, and the highest

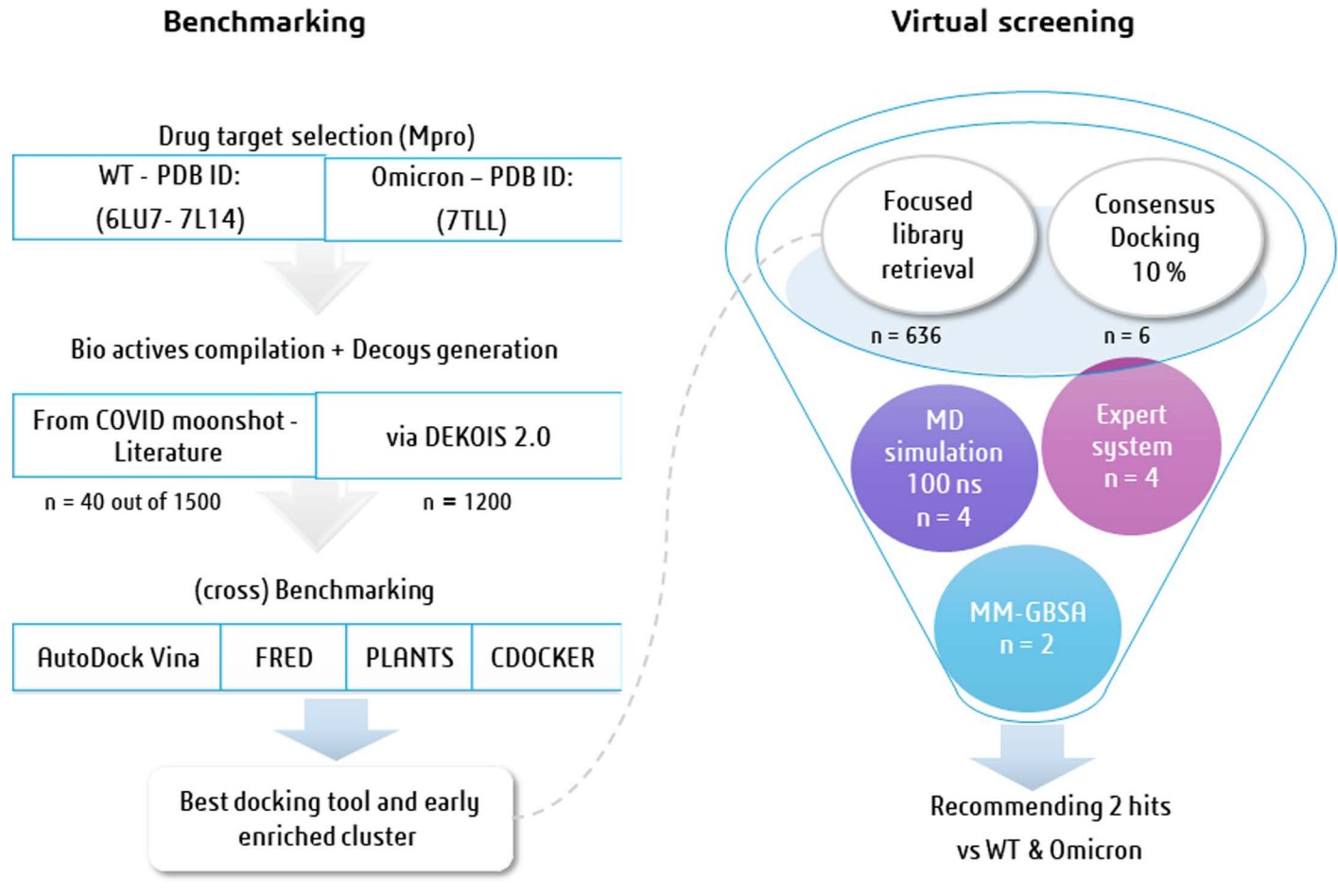

**Fig 1. Logical flow of the planned work.**

similarity fell below 0.6. DEKOIS 2.0 protocol [18,19] was employed to the compiled 40 structures of Mpro bioactives retrieved from both COVID-Moonshot and literature to create 1200 decoys (1:30 ratio) [24–26].

## Preparation of small molecules

All the small molecules were built and prepared using the Molecular Operating Environment (MOE) version 2019.01 [27]. The preparation procedure was applied as reported earlier [28]. More details can be found in S1 File.

The SD files were transformed and split into separate PDBQT files for docking studies with AutoDock Vina using OpenBabel [29]. For PLANTS, the molecules were converted to mol2 format and the compatible atom types were specified using SPORES software [30,31].

## Preparation of the protein structures

Prior to the docking studies, the protein structure of SARS-CoV-2 WTMpro (PDB ID: 6LU7 and 7L14) and OMPro (PDB ID: 7TLL) were prepared using MOE, as reported earlier [28]. Further preparation details can be found in the S1 File.

## Screening and docking experiments

For screening using AutoDock Vina (version 1.1.2) [32], the Python script (*prepare receptor4. py*) offered by the MGLTools package (version 1.5.4) [33] was used to convert protein files to PDBQT format. The efficiency of the search algorithm was retained at its default settings. The grid box docking dimensions were 22.88 Å × 29.62 Å × 25.12 Å, with a spacing of 1 Å to deal with all the possible conformations of the docked molecules.

For screening using PLANTS, "ChemPLP" was used as the scoring function for docking utilizing PLANTS [34] with the "screen" mode selected. The receptor atoms around the co-crystal ligand coordinates by 5 Å were included in the binding site. Then the best-scored pose of each ligand was retrieved for further analysis.

For screening using FRED, Multi-conformer file generation for the small molecule database was carried out using the OMEGA module of OpenEye software [35] with a default number of 200 conformers generated for each ligand. The protein structures were prepared using MakeReceptor 4.1.1.0 GUI module of OpenEye software. Virtual screening was carried out using FRED module of OpenEye software [36,37].

For the screening using CDOCKER [38], a grid-based docking program (version 5.5). The binding site sphere of Mpro was defined as the regions that come within 14 Å radius from the geometric centroid of the co-crystalized ligand to make sure that all geometries of the docked compounds are covered. Other docking parameters were kept at the default level. The same docking protocol was carried out for WTMpro and OMpro sequences of SARS-COV-2.

## Benchmarking utilizing pROC calculations and pROC-Chemotype plotting

The score-ordered rank was used in calculating the pROC-AUC using "R-Snippet" component of KNIME [39], based on the following equation [40]:

$$pROC\ AUC = \frac{1}{n} \sum_{i}^{n} \left[ -log_{10}(D_i) \right] = \frac{1}{n} \sum_{i}^{n} log_{10} \left( \frac{1}{D_i} \right)$$

The bioactives number is given by *n*, while *Di* is the decoys fraction that is ordered higher than *ith* bioactive detected. The *ith* is the number of bioactive in the rank.

The pROC-Chemotype plots were plotted by the "pROC-Chemotype plot" tool [41].

To assess the ability of the docking tool to recognize true positives, from the active set, in the score-ordered list compared to the random collection, enrichment factor (EF) was computed based on the following equation [42] :

$$EF = \frac{\dfrac{Bioactives_{subset}}{N_{subset}}}{\dfrac{Bioactives_{total}}{N_{total}}}$$

## Drug likeness and ADME/TOX

Drug likeness and ADMET date were checked using SwissADME (http://www.swissadme. ch/) after generation of SMILES for the compounds using mol files using the pkCSM webtool (https://biosig.lab.uq.edu.au/pkcsm/).

## Molecular dynamics simulations

Molecular dynamics simulations were conducted using GROMACS 2022 [43]. The solvation of each protein-ligand complex was carried out in a dodecahedron box of TIP3P explicit water

model [44]. The preparation, equilibration and production settings were adapted as reported earlier [28], with an exception that the production runs were for 100 ns. Detailed procedures can be found in the S1 File.

The trajectories obtained from the MD simulations were analyzed via RMSD, RMSF, RoG, H-bond count and minimum distance analysis, using tools available in the GROMACS 2022 package. Graphical representations were made using Grace Software (https://plasma-gate. weizmann.ac.il/Grace/).

### MM GBSA calculations

The binding free energy of the top hits and reference compounds were assessed via MM GBSA approach by applying gmx_MMPBSA tool v1.5.2 [45]. The tool was conjugated with GROMACS to conduct AMBER's based calculations [46].

## Results and discussion

### Structural insights on Mpro

Mpro is a homodimer composed of three domains, the *N*-terminal domain I (residues 10–99), domain II (residues 100–182), where both presenting the catalytic domains, and the C-terminal helical domain III (residues 198–303) [12], as illustrated in **Fig 2**. Mpro binding site (**Fig 2B**) constructs vital subsites near the cleavage region of the substrate, named S1, S2, S3, S4 and S5 [11]. They correspond to the positions P1, P2, P1′ that are essential for substrate specificity, and P3, P4 which are essential for substrate binding stability (**Fig 2C**) [11]. S1 holds the oxyanion hole loop (Gly138 – Gly146), while S2 holds a catalytic dyad between the interface of domain I with HIS41 and CYS145 from domain II [47]. The catalytic loop is formed via residues Ile43 – Tyr54 together with the residues Glu166–His172 from P1 position [47]. Finally, the S4 linker loop is composed of residues Asp187 – Ala193 [47].

Although the OMpro P132H mutation is not directly at Mpro's catalytic site and being 22 Å away (**Fig 2A** and B)**,** it was found to cause resistance to the therapeutic effects of drugs [48]. A recent comparative molecular dynamics (MD) study shed light on the P132H mutation resistance mechanism proposing the possibility of this mutation's ability to alter the enzyme's conformational flexibility [49], despite an unchanged catalytic efficiency and a compromised thermal stability that was previously reported [48].

In this study, we used the Mpro structure (PDB ID: 6LU7) in our benchmarking analysis, VS and MD studies since it was frequently employed in different studies [26,50,51]. Furthermore, we selected another WT structure (PDB ID: 7L14) for the VS and MD investigation since it reflects a non-covalent binding scenario for small molecules to Mpro. Moreover, the mutant OMPro P132H (PDB ID: 7TLL) was utilized for the cross-benchmarking investigations, and subsequent VS and MD experiments.

### Selection of the active set molecules

To build our actives set, we downloaded Mpro inhibitors that were available at the COVID-Moonshot website (https://dndi.org/research-development/portfolio/covid-moonshot/). Of those compounds, we excluded all compounds with covalent warheads and fragments. Additionally, we manually collected 35 non-peptide, non-covalent compounds from the literature [24–26] to attain a maximum diversity in the chemotype clusters. Subsequently, we merged the inhibitors, and filtration was carried out to exclude ligands with $IC_{50}$ values above 2 µM or containing chiral centers. At the end, we selected 40 bioactive compounds comprising 7 co-crystal ligands with Mpro protein structures, with PDB codes: 7L13, 7M8P,

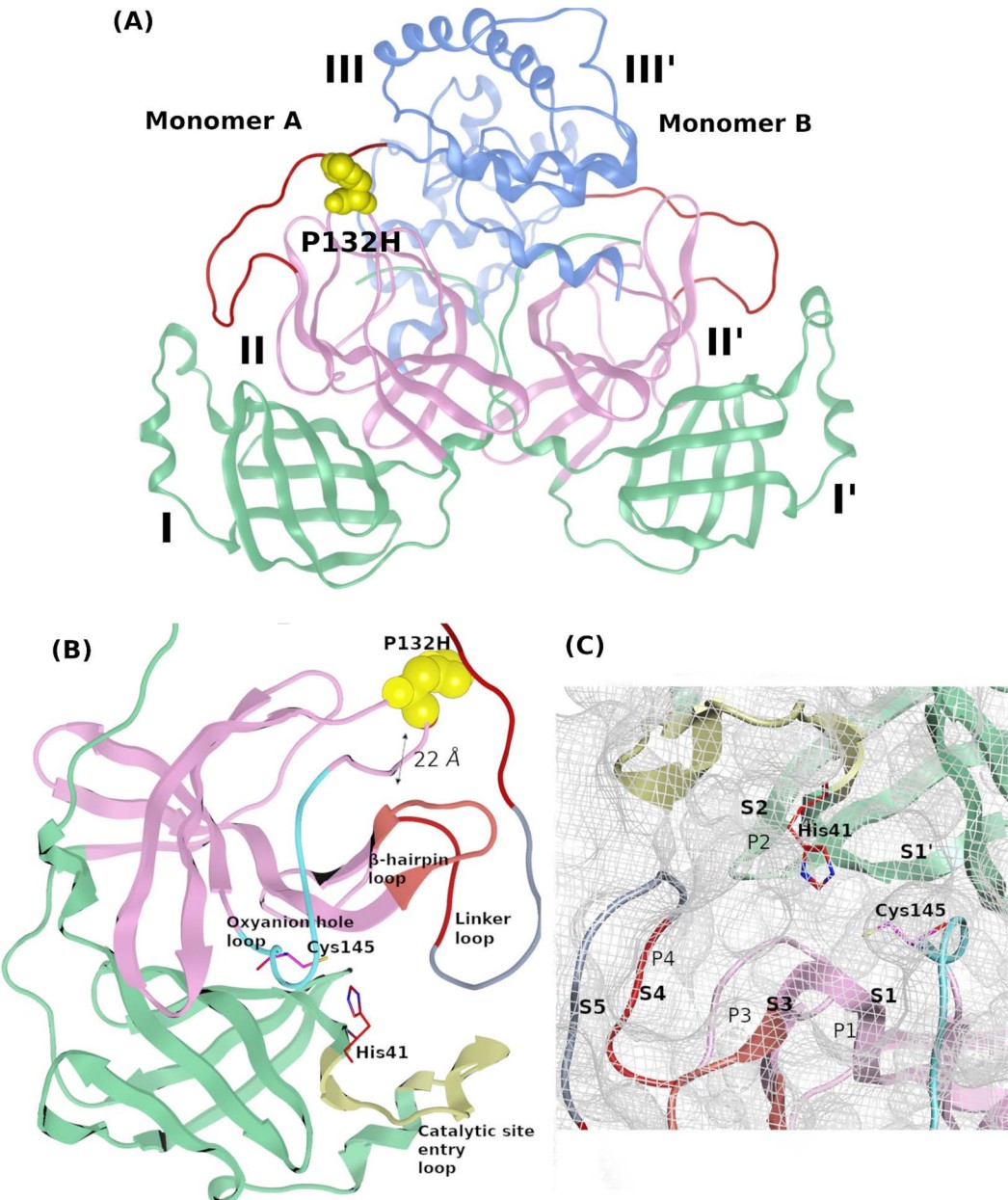

**Fig 2. The crystal structure of dimer SARS-CoV-2 Mpro Omicron mutant P132H (PDB ID 7TLL).** (A) The three different domains I, II, and III of the protein structure. (B) Structure of the catalytic pocket loops showing the site of P132H mutation; (C) the substrate-binding subpockets of Mpro.

7M91, 7M8N, 7L11, 7L14, and 6M2N. Upon clustering, these compounds were found to collectively represent 19 scaffold chemotype classes as presented in **Table A in** S1 File. The complete list of the 40 bioactive compounds can be found in **Table B in** S1 File. For decoys generation, these 40 candidates were subjected to the DEKOIS 2.0 protocol [18,19] which creates 30 structurally related decoys per compound. Eventually, we compiled a challenging decoy set of 1200 compounds where the whole set of bioactives and decoys was used to assess the performance of the four docking tools investigated.

## Benchmarking analysis

A docking study of the 1240 actives plus decoys was launched on the four docking tools AutoDock Vina, FRED, PLANTS, and CDOCKER to assess their performance. The results obtained from this docking investigation reflected that all docking tools exhibited better-than-random performance judged by having a pROC-AUC value of more than 0.43 for the two assessed Mpro sequences. Noteworthy is that pROC-AUC is a modified version plotted in semi-logarithmic scale instead of a linear receiver operating characteristics (ROC) curve [40]. Therefore, the calculation of the area under the curve (AUC) is biased to emphasize the detection of early enriched actives [40]. For the WTMpro sequence (PDB ID: 6LU7), AutoDock Vina showed the best screening performance with pROC-AUC value of 0.98, followed by FRED with pROC-AUC value of 0.68, as seen in (**Fig 3**). However, cross-benchmarking experiments for the OMPro mutant recommended both FRED and AutoDock Vina for the best screening performance by virtue of their pROC-AUC values of 1.2, and 0.92, respectively (**Fig 3**). Collectively, AutoDock Vina and FRED showed superior predictive power compared to the other two tools. This directed us to utilize AutoDock Vina in the subsequent experimental explorations adopted in this study.

## Chemotype enrichment

To investigate the highest-ranking cluster/chemotype, an in-depth performance analysis *via* pROC-Chemotype plots was conducted [41]. The results illustrate a high enrichment of the best potent bioactives in the early docking ranks. This was done for the docking results obtained from AutoDock Vina against both the WT and Omicron Mpro structures and FRED against Omicron Mpro, in Figs 4–6, respectively. It is worth mentioning that there is

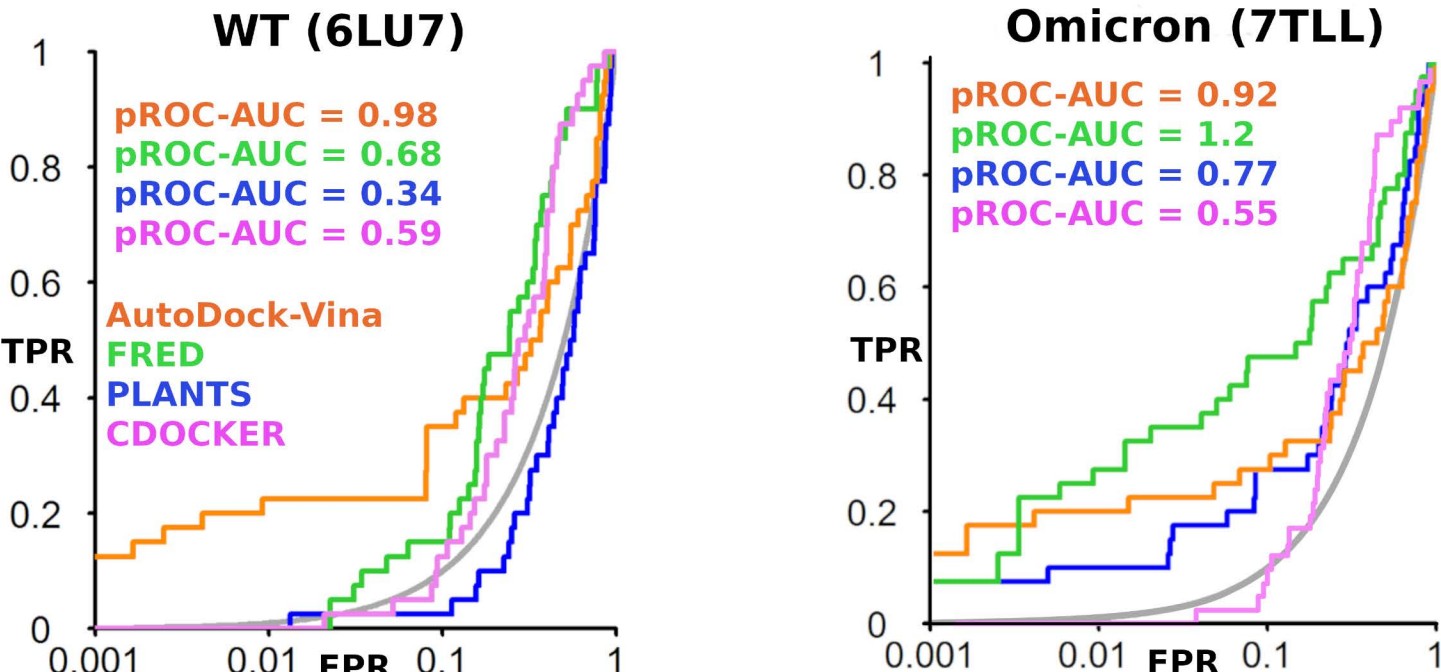

**Fig 3. pROC plots illustrating the screening performance of AutoDock Vina, FRED, PLANTS and CDOCKER.** The left panel is for the SARS-CoV-2 WTMpro (PDB ID: 6LU7). The right panel is for SARS-CoV-2 OMpro P132H mutant.

no reported data available about the inhibitory effect of the active set compounds against the OMpro variant. Therefore, the cross-benchmarking experiment against the OMpro using the benchmark set of WTMpro can provide insights on how the docking tool behave against a closely similar protein structure with completely identical binding site residues.

The diversity of the bioactive chemotype clusters (19 distinct scaffolds) underlines the challenge inherent in evaluating the performance of the utilized docking tools. Their bioactivity profiles are symbolized by the level of activity (LOA), extending from $10^{-6}$ to $10^{-8}$ M, and reported as $IC_{50}$ as a type of data (TOD), as displayed in (**Fig 4A**). The pROC-Chemotype plot visualized that AutoDock Vina can identify potent binders at early enrichment. For instance, at 1% of the score-ordered library, many bioactive molecules were recognized, leading to an Enrichment Factor (EF 1%) of 18.08 in case of the WTMpro (**Fig 4A**), especially at 1% of the score-ordered library. Furthermore, the best four potent bioactive compounds (with the bioactivity rank = 1-4) were recognized at the same cutoff. The enrichment factor (EF 1%) is 18.08, emphasizing a promising predictive capability of the docking tool. This signifies that AutoDock Vina with WTMpro is able to recognize active molecules more than the random performance by 18-fold at early enrichment. Furthermore, **Fig 4B** exhibits the docking fitness (fitness = docking score multiplied by −1) allocation of the bioactive molecules. The docking score is ranging from −9.8 (best score) to −6.7 (lowest score) and presented as fitness values of 9.8 to 6.7. Furthermore, molecules demonstrating cluster 1 were found to lie in a superior area of fitness (i.e., fitness > 9) compared to other scaffolds (**Fig 4A**). Docking pose of the best ranked compound (docking rank 1) from the active set in the binding site (PDB ID: 6LU7) is shown in **Fig 4C**, where key interactions with residues; His163, Cys145, Thr26 and His41 were mapped.

For the cross-benchmarking against OMpro, both FRED and AutoDock Vina exhibited the best performance with pROC-AUC values of 1.2 and 0.92, respectively. FRED chemotype analysis indicated an EF 1% = 20.4 (**Fig 5A**). The fitness distribution of the bioactive molecules exhibited that scaffolds of cluster 6 and 7 were best recognized at the early rank, as shown in **Fig 5B**. The best-scored docking pose (compound **12**) presented the key interactions with Glu166, His163 and Phe140 (**Fig 5C**).

Unlike FRED, AutoDock Vina was able to enrich the most potent active (compound **1**, bioactivity rank = 1) as the best scored docking pose (docking rank = 1), as well as the subsequent four potent molecules (bioactivity rank = 2-5), as shown in **Fig 6A**. Notably, scaffold of cluster 1 is entirely enriched at the early enrichment (at 1%), as revealed in the fitness distributions of the clusters in **Fig 6B**. Elucidating the interactions of the best ranked docking pose in the binding site of OMPro, compound **1** presented the key interactions with Cys145, His163 and Thr26 (**Fig 6C**).

## Virtual screening of a focused library based on benchmarking results

Based on the chemotype behavior in Figs 4 and 6, the most potent bioactive cluster (cluster 1 -N-aryl pyridone derivatives) were retrieved by AutoDock Vina at early enrichment. Interestingly, the literature reported that perampanel derivatives (cluster 1 scaffold) display a promising skeleton for novel inhibitors for WTMpro [26,52]. Therefore, this top enriched chemotype class encouraged us to integrate the parent drug (perampanel) into a scaffold hopping transformation strategy using the DrugspaceX database https://drugspacex.simm. ac.cn/ that enables expert-defined transformations of approved drugs for virtual screening with lingering structural novelty. We hence generated a focused library of 636 compounds based on class 1 scaffold for prospective virtual screening against the WTMpro. Furthermore, we selected a non-covalent co-crystal structure (PDB ID: 7L14) for the VS effort complexed with compound number 9 (cluster 1) in **Table B in** S1 File. It is noteworthy that upon

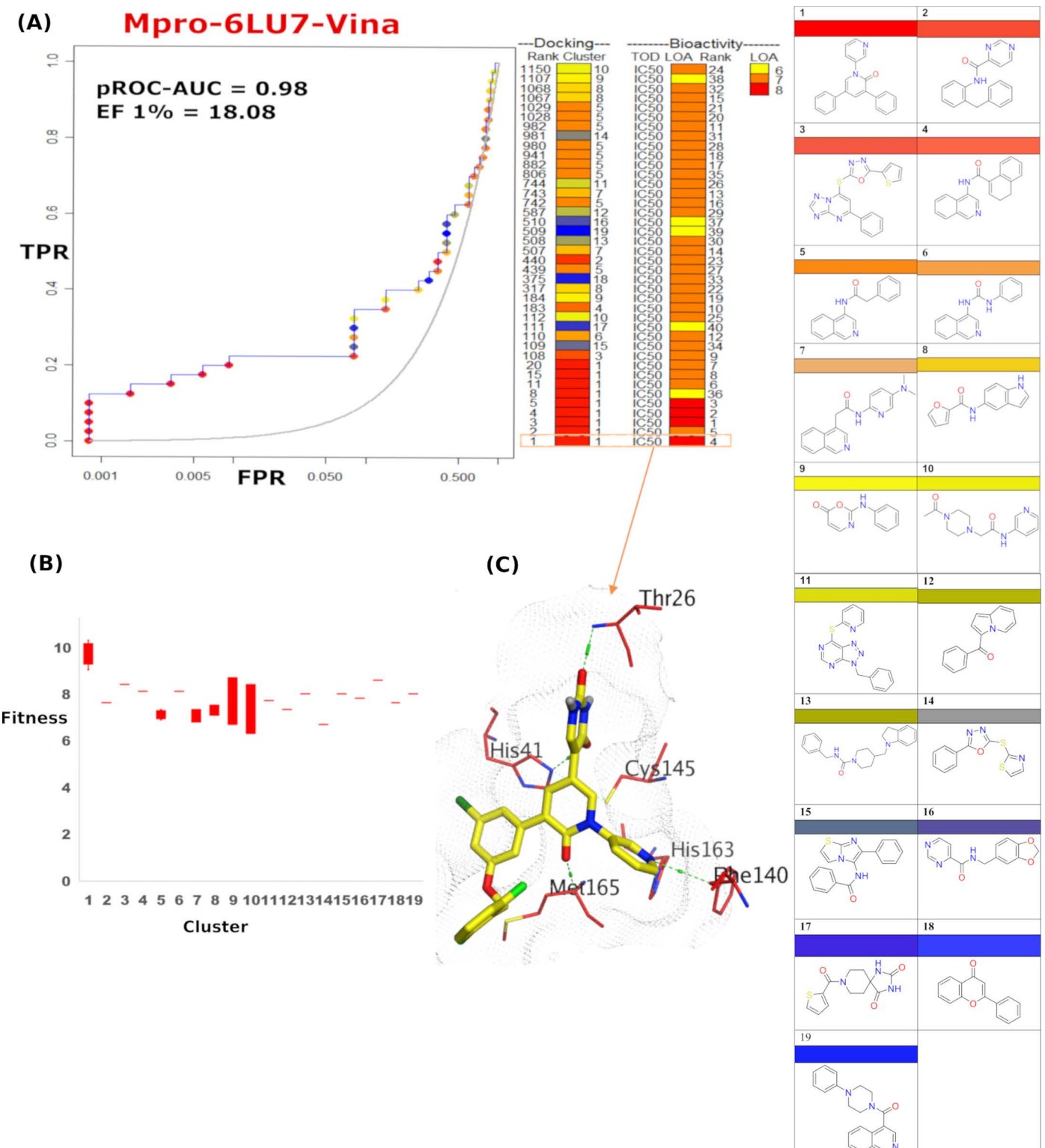

**Fig 4. pROC-Chemotype plot for the benchmarking using AutoDock Vina against SARS-CoV-2 WTMpro.** (A) The pROC-Chemotype plot where the docking information corresponds to the chemotype described by the cluster number and the bioactivity information. (B) Box plot of the fitness vs. chemotype clusters. (C) 3D representation of the best ranked bioactive molecule in the binding site of Mpro.

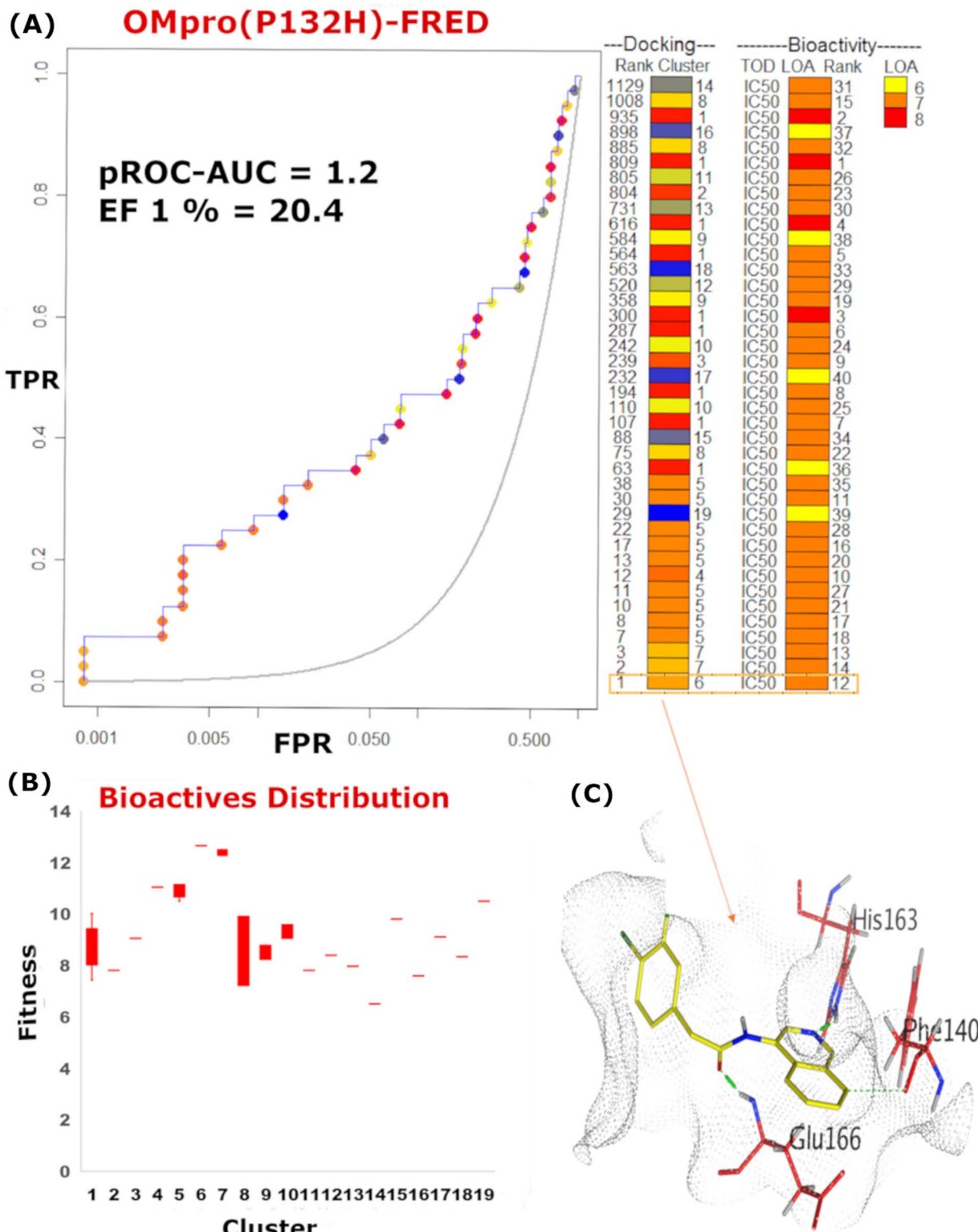

**Fig 5. pROC-Chemotype plot for the benchmarking using FRED against SARS-CoV-2 OMpro (P132H).** (A) The pROC-Chemotype plot where the docking information corresponds to the chemotype described by the cluster number and the bioactivity information. (B) Box plot of the fitness vs. chemotype clusters. (C) 3D representation of the best ranked bioactive molecule in the binding site of Mpro.

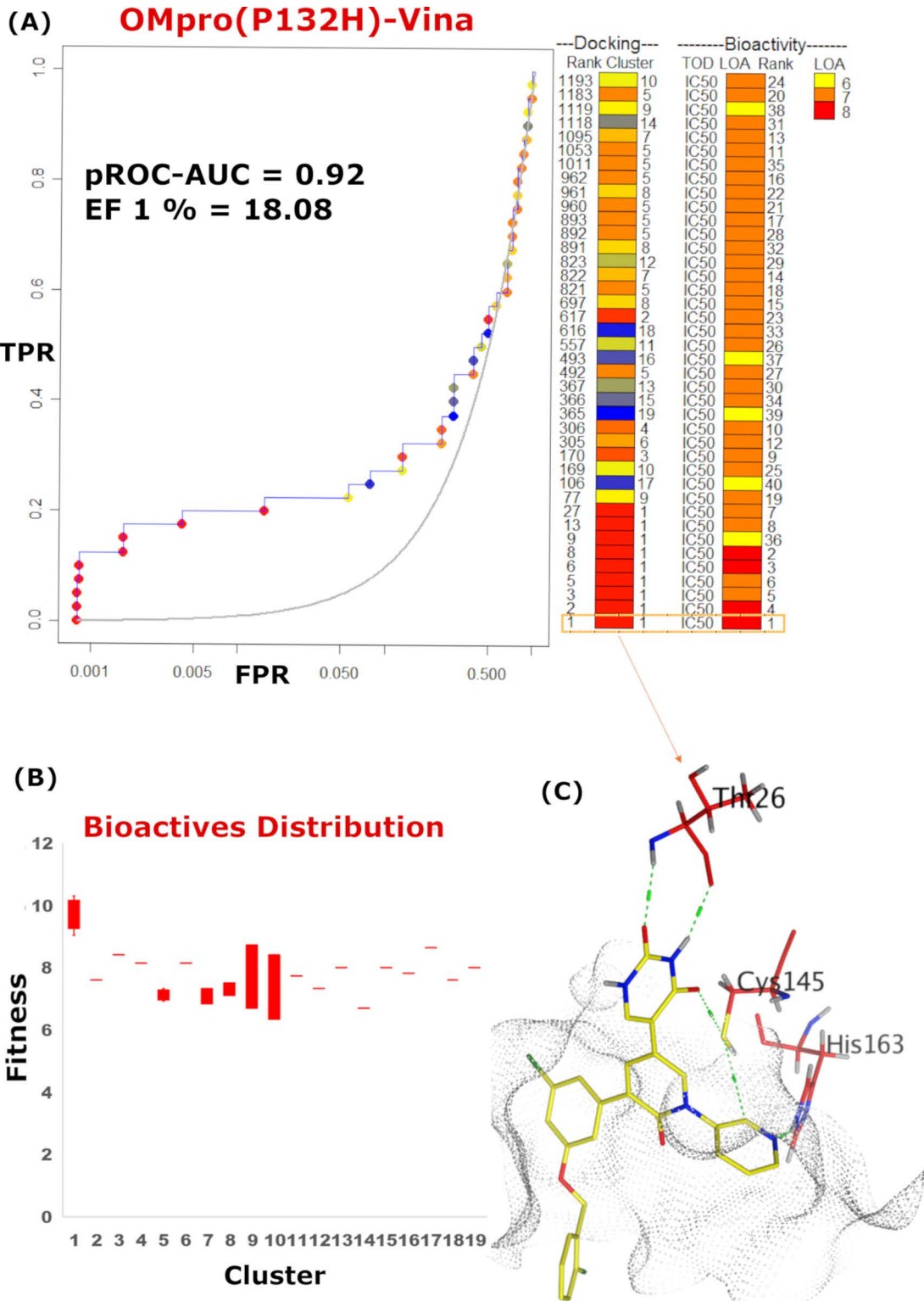

**Fig 6. pROC-Chemotype plot for the benchmarking using AutoDock Vina against SARS-CoV-2 OMpro (P132H).** (A) The pROC-Chemotype plot where the docking information corresponds to the chemotype described by the cluster number and the bioactivity information. (B) Box plot of the fitness vs. chemotype clusters. (C) 3D representation of the best ranked bioactive molecule in the binding site of Mpro.

docking of the 3 top ranked compounds against 6LU7on 7L14, we obtained similar docking ranks where compound **203** came first, then compound **386**, then compound **329**, as shown in (Table 1).

The generated library was screened against OMpro (PDB ID: 7TLL) complexed with nirmatrelvir using both FRED and AutoDock Vina. We focused on the top 10% of the best scored compounds using both docking tools and explored the consensus between both of them. Moreover, focusing on detecting the compounds with comparable poses and similar important interactions, we were able to identify 3 compounds as best potential ligands for OMpro *viz.* compound **401** with scores of −9.4, −12.4; compound **541** with scores of −9.4, and −10.6; and compound **385** with scores of −10.15 and −9.9 for AutoDock Vina and FRED, respectively (**Table 1** and **Table C in** S1 File).

## Description of the postulated binding poses

The docking pose of compound **386**, transformed from perampanel by replacing the phenyl ring with a bioisosteric tetrazole ring, showed multiple interactions including five anticipated hydrogen bonds between the ligand and the WTMpro (**Fig 7A**). The pyridone carbonyl oxygen showed two H-bonding interactions with Gly143 and Asn142, and a hydrophobic contact with Cys145. The cyanophenyl ring displayed a hydrophobic contact with Glu166 while forming H-bonding interaction with Cys145. The tetrazole ring formed a H-bonding interaction with the backbone of Thr26. Finally, the pyridine ring showed hydrophobic interaction with Met165 of S4 pocket.

The pose of compound **329**, transformed from the perampanel by changing the phenyl ring to hydroxytriazoline, demonstrated good binding with the active site of WTMpro. The hydroxytriazoline group formed H-bonding interaction with Ser46, and the pyridone carbonyl oxygen exhibited also H-bonding interactions with Gly143 and Asn142. Additionally, Cys145 was involved in a hydrophobic contact with the same ring (**Fig 7B**).

The docking pose of compound **203** (**Fig 7C**) showed that the chromeno pyridine group established three H-bonding interactions with Glu166 (P1 catalytic loop), Gln189 (S4 sub pocket) and Thr25 (S1' pocket).

**Table 1. The best enriched compounds against WTMpro and OMpro.**

| Compound ID[a] | Vina Score (Kcal/mol) | FRED Score | M. Wt. | Mpro Target |
|---|---|---|---|---|
| **Compound 386** | −9.5 −10.7 | – | 447.45 | WT (7L14) (6LU7) |
| **Compound 329** | −9.3 −10.2 | – | 490.47 | WT (7L14) (6LU7) |
| **Compound 203** | −10.7 −10.9 | – | 468.46 | WT (7L14) (6LU7) |
| **Compound 401** | −9.6 | −12.4 | 418.49 | Omicron (P132H) |
| **Compound 541** | −9.2 | −10.6 | 388.42 | Omicron (P132H) |
| **Compound 385** | −10.15 | −9.9 | 468.46 | Omicron (P132H) |

[a]The structures of the compounds can be found in **Table C in** S1 File.

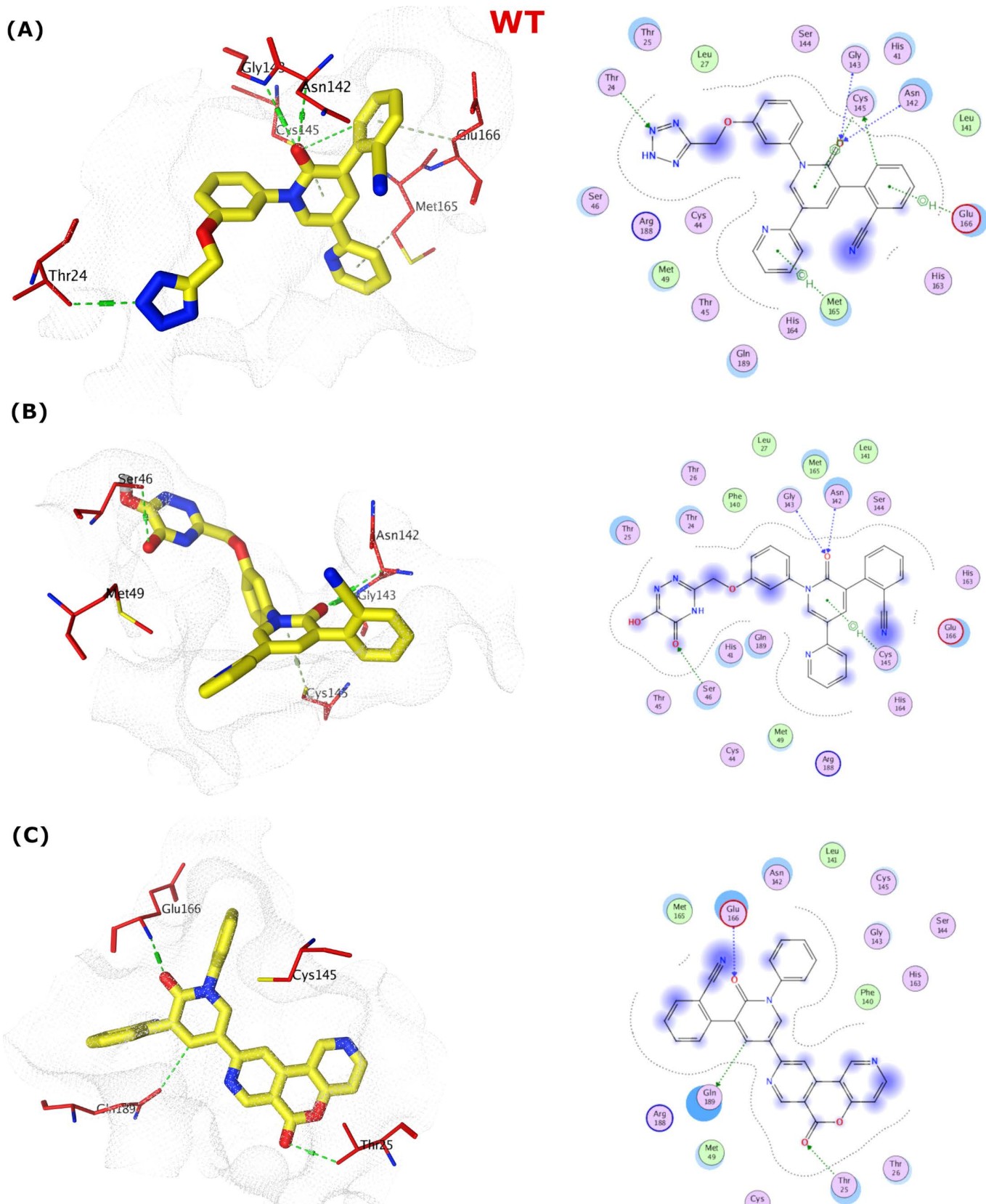

**Fig 7. The docking poses in binding site of the WTMpro (PDB ID: 7L14) represented by 3D and 2D.** (A) The pose of compound **386**. (B) The pose of compound **329**. (C) The pose of compound **203**.

The postulated binding pose of compound **401** exhibited three H-bond interactions with Cys145, His163 and Glu166 (S1 pocket). Furthermore, favorable interactions with Met49 (S2 pocket), and Met165 and Gly143 (S4 pocket) were observed (**Fig 8A**).

The pose of compound **541** showed four H-bonding interactions with Thr26, Met49 (S2 pocket), and His163, Cys145 (S1 pocket), as shown in **Fig 8B** [53]. Finally, the pose of compound **385** displayed H-bonding interaction with Cys145, Glu166 and Met165 (**Fig 8C**).

## ADME/Tox and drug-likeness filtration applying expert systems

To assess the pharmacokinetic (PK) profile of the top hits, pkCSM (http://biosig.unimelb.edu.au/ pkcsm/) online server was used. Starting with the absorption parameter as represented by the Intestinal absorption parameter, the value of which must be greater than 30%, compounds **203**, **401**, **541** and **385** showed 100% values, while compounds **386** and **329** showed 93% and 72% values, respectively. For BBB permeability, if a compound has a log BB value greater than 0.3, it is anticipated to readily cross the BBB, while if the value is less than −1, it will poorly be distributed to the brain. For the evaluated compounds, only compounds **401**, and **541** were found to demonstrate possible BBB permeation ability. This observation predicts the lack of potential CNS side effects of the majority of the compounds. The toxicity prediction showed that only compound **203** has positive AMES toxicity (**Table 2**). To evaluate the drug-likeness and PK profile of the compounds we used Swiss ADME (http://www.swissadme.ch/). Interestingly, no compound showed a violation of Lipinski's rule. All the data related to the number of H bond donors and acceptors, consensus Log Po/w, TPSA, and bioavailability are shown in (**Table 3**). Collectively, all compounds carried acceptable criteria, except compounds **203**, and **385**. Therefore, we excluded those two compounds from further investigation due to their potential toxicity, and poor solubility profiles.

## Molecular dynamics simulations

The two top-enriched ligands from DrugSpaceX for each of the WTMpro (compounds **386** and **329**) and OMpro (compounds **401** and **541)** were exposed to 100 ns molecular dynamics (MD) simulations to assess their stability in the binding sites in a time-dependent manner. Furthermore, for comparison purposes, additional MD runs were conducted for the non-liganded (apo) proteins and co-crystallized systems for both PDB ID 7L14, and 7TLL pertaining to the WTMpro and OMpro, respectively. This results in a total of eight MD runs, with a 100 ns each.

## Root mean square deviation (RMSD)

For the WTMpro complex systems, the root mean square deviation (RMSD) for ligand's heavy atoms showed high fluctuation between 2-3 nm for the co-crystallized ligand (blue). While both compounds **386** and **329** exhibited stable fluctuation with RMSD less than 0.5 nm (**Fig 9A**). The stability of the protein dynamics was evaluated via RMSD which was calculated on the alpha carbon atoms. At the first 20 ns; the apoprotein (black) exhibited RMSD values between 0.1 to 0.4 nm. Compound **329** (green) was able to stabilize the protein structure showing an RMSD value of less than 0.2 nm, then exhibited a higher fluctuation around an RMSD value of 0.3 nm. Likewise, the co-crystal ligand (blue) and compound **386** (red) showed acceptable RMSD values around 0.3 nm especially after 20 ns of time until the end of 100 ns simulation time (**Fig 9B**). Intermolecular hydrogen bonding can be utilized as a metric to evaluate the degree of protein-ligand binding as well as the stability of the complex. During a 100 ns simulation, the total number of hydrogen bonds formed by the two hit compounds ranged from 1 to 8, compared to a maximum of 4 H-bonding count for the co-crystal ligand, as depicted in (**Fig 9C**). This indicates the potential stable binding of these compounds.

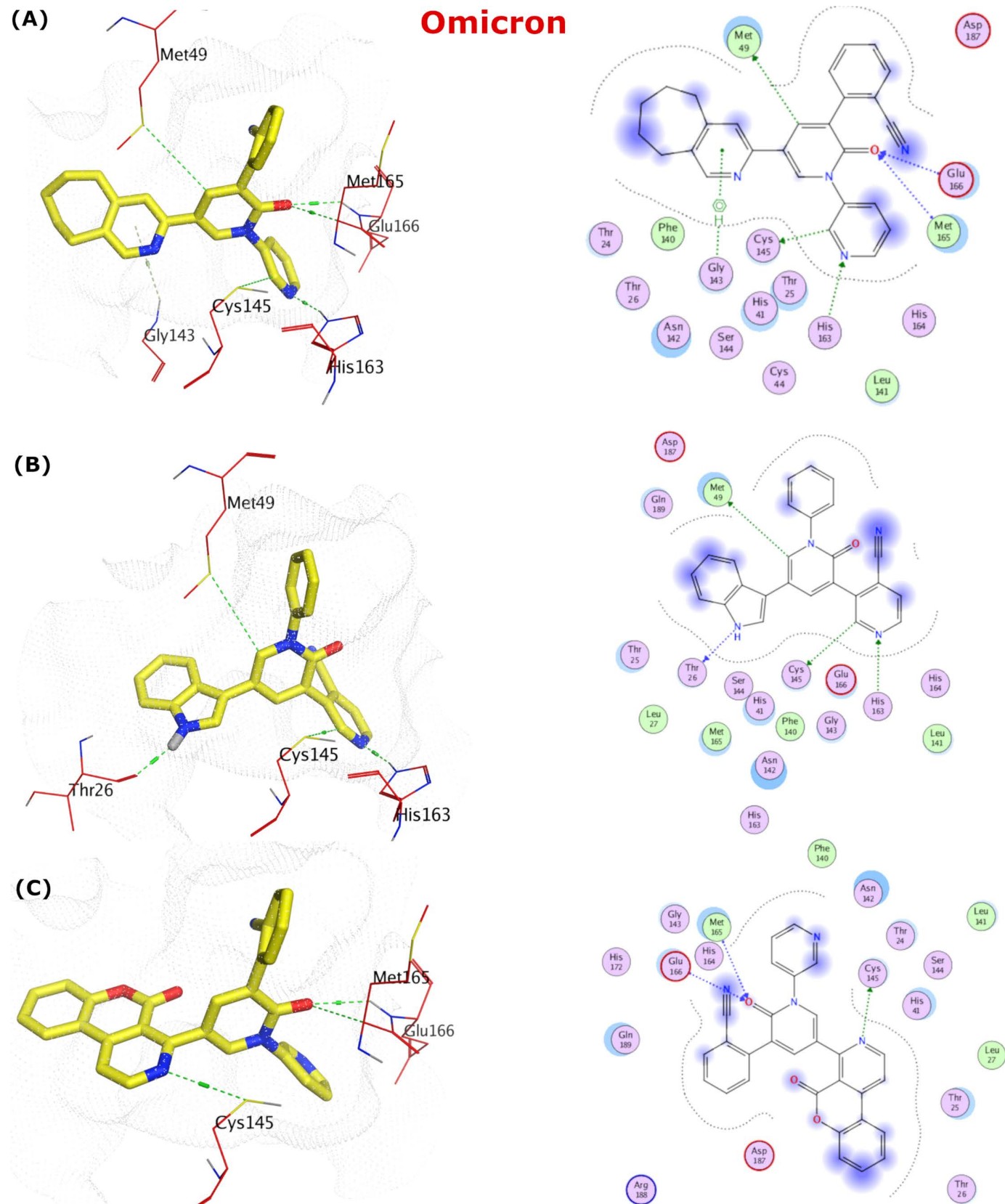

**Fig 8. The docking poses in binding site of the OMpro (PDB ID: 7TLL) represented by 3D and 2D.** (A) The pose of compound **401**. (B) The pose of compound **541**. (C) The pose of compound **385**.

**Table 2. ADME/Tox data for hits against WTMpro and OMpro.**

| Compound # | Intestinal absorption (% Absorbed) | BBB permeability (log BB) | AMES toxicity |
|---|---|---|---|
| 386 | 93.64 | −1.457 | No |
| 329 | 72.98 | −1.81 | No |
| 203 | 100 | −1.055 | Yes |
| 401 | 100 | −0.503 | No |
| 541 | 100 | −0.231 | No |
| 385 | 100 | −1.043 | No |

**Table 3. Drug likeness, Lipinski's rule, and PK properties for hits against WTMpro and OMpro.**

| Compound # | Num. H-bond | | Log Po/w (Consensus) | Δ TPSA (Å²) | Drug likeness (Lipinski's Rule of 5) | | Bioavail-ability | Solubility |
|---|---|---|---|---|---|---|---|---|
| | Acceptors | Donors | | | Follow | Violation | | |
| 386 | 8 | 0 | 2.79 | 119.47 | Yes | 0 | 0.56 | Moderately soluble |
| 329 | 8 | 2 | 2.78 | 146.78 | Yes | 0 | 0.55 | Moderately soluble |
| 203 | 6 | 0 | 3.99 | 101.78 | Yes | 0 | 0.55 | Insoluble |
| 401 | 4 | 0 | 4.3 | 71.57 | Yes | 0 | 0.55 | Moderately soluble |
| 541 | 3 | 1 | 3.8 | −5.6 | Yes | 0 | 0.55 | Moderately soluble |
| 385 | 6 | 0 | 3.95 | 101.78 | Yes | 0 | 0.55 | Insoluble |

For OMpro complex systems, the ligand's heavy atoms RMSD was evaluated. At the first 40 ns the co-crystallized ligand showed fluctuations of the RMSD values between 0.1-0.5 nm then stabilized between 0.1-0.3 nm afterwards. Similarly, compounds **401** and **541** showed fluctuations around RMSD values 0.1-0.3 nm then 0.2-0.35 nm till the end of the simulation (**Fig 10A**). Assessing the protein dynamics stability via RMSD (**Fig 10B**) indicated acceptable fluctuations for all systems. For instance, RMSD values showed a range from 0.2 to 0.4 nm of co-crystal complex (blue) and compound **401** (red), and from 0.1 to 0.3 nm for compound **541** (green). Furthermore, the unliganded (apo) showed minor fluctuations ranging from RMSD values of 0.2 to 0.3 nm. The H-bond count for the complex systems (**Fig 10C**) showed good stability for both compound **401** (red) and compound **541** (green), with at least 2 to 3 H-bond count along the simulation time. The co-crystal ligand exhibited 4 H-bond count on average, especially after 40 ns of simulation time.

## Radius of gyration

The Radius of gyration (RoG) is a general metric for the compactness of the protein during the simulation time. The WT-complex systems displayed acceptable RoG value ranges for all systems. For instance, the unliganded system (black) exhibited RoG values from 2.2 to 2.3 nm (**Fig 11A**). Likewise, co-crystal (blue), compound **386** (red) and compound **329** (green) complex systems showed a narrow range of RoG values, from 2.2 to 2.25 nm. This indicates the acceptable compactness of the simulated systems and the absence of major conformational discrepancies. The same outcome can be concluded from the OMpro complex systems (**Fig 11C**).

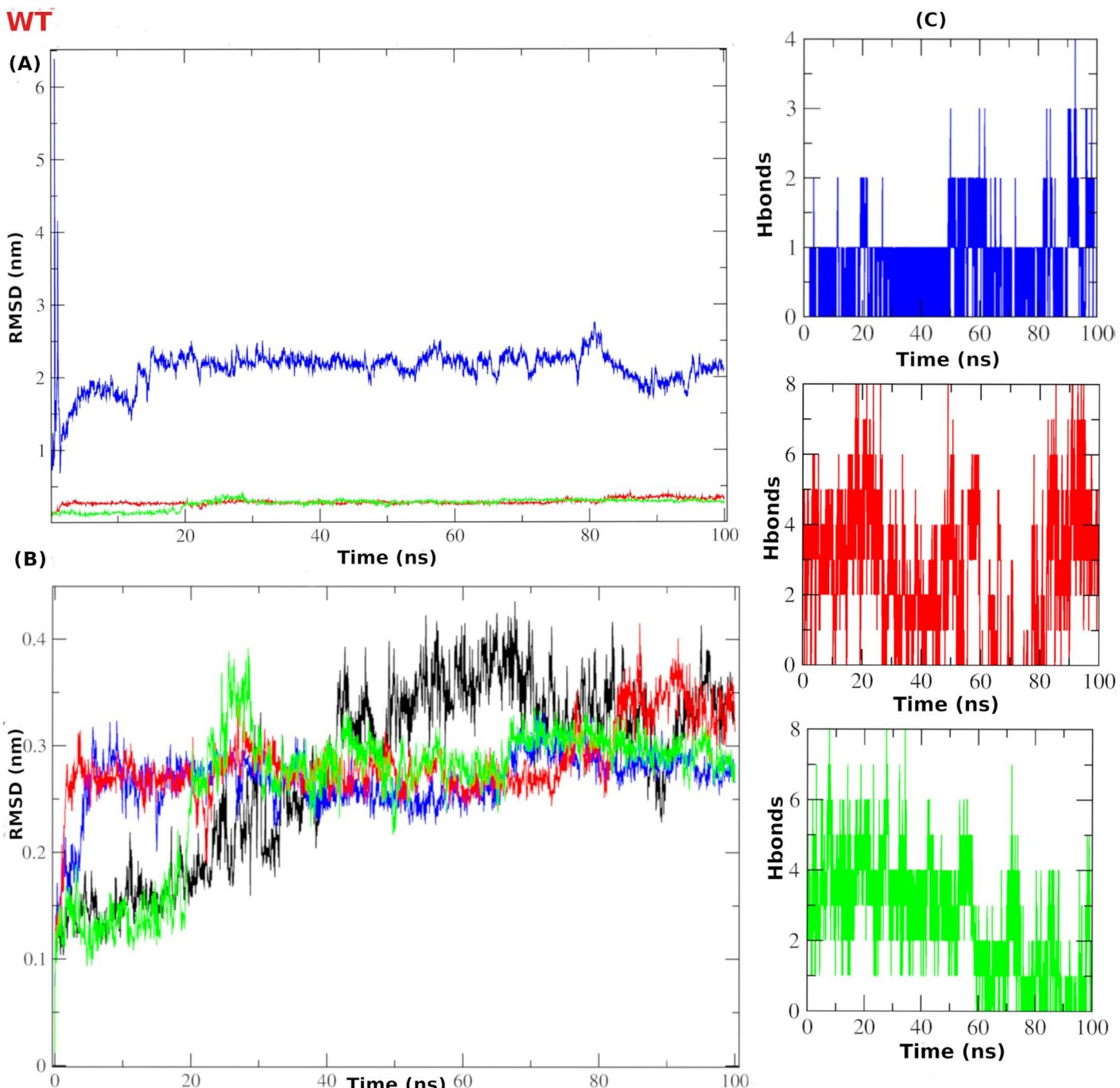

**Fig 9. The RMSD and H-bonding count analysis of the MD simulation for the WTMpro complex systems.** (A) Ligand RMSD. (B) Protein RMSD. (C) H-bonding count. Blue line is for co-crystal ligand, while red is for compound **386** and green is for compound **329**, in all graphs.

### RMSF analysis

Root mean square fluctuation (RMSF) quantifies the conformational perturbations per-residue during the simulation time. We noticed shared flexibility behavior in all systems in certain regions, for the complex systems with WTMpro (**Fig 11B**), and the complex systems

# Omicron

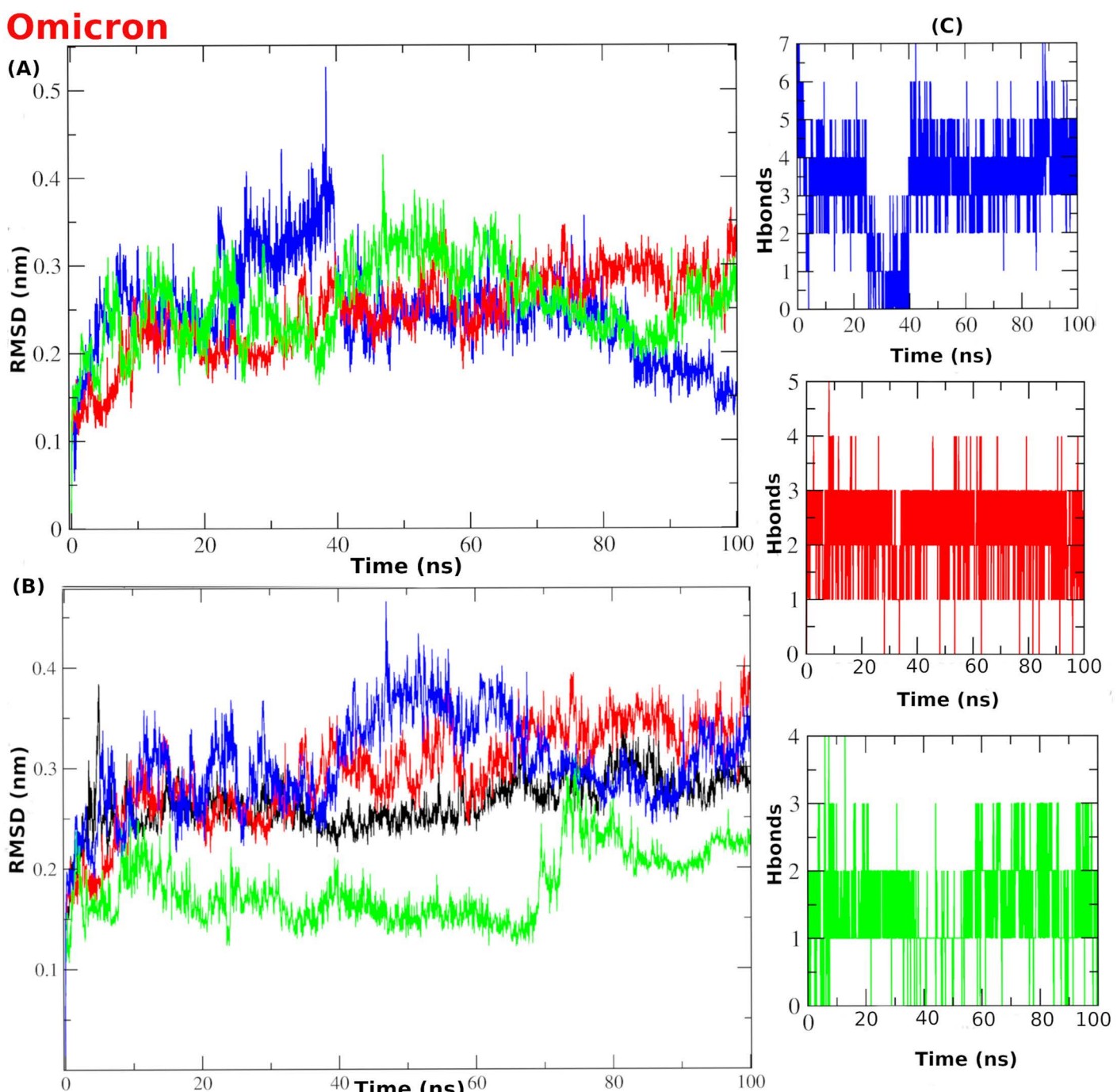

**Fig 10. The RMSD and H-bonding count analysis of the MD simulation for the OMpro complex systems.** (A) Ligand RMSD. (B) Protein RMSD. (C) H-bonding count. Blue line is for the co-crystal ligand, while red is for compound **401** and green is for compound **541**, in all graphs.

with the OMpro (**Fig 11D**). For Instance, the P132H mutation region of OMpro, the co-crystal and compound **401** showed more stable fluctuation than the apo structure. Comparatively, both showed fluctuation lower than 0.2 nm except in residues around 225 and 275. However, compound **451**- OMpro (green) displayed high flexibility in most regions. Some high fluctuations in the RMSF values are reflected in the flexibility of structural loop regions

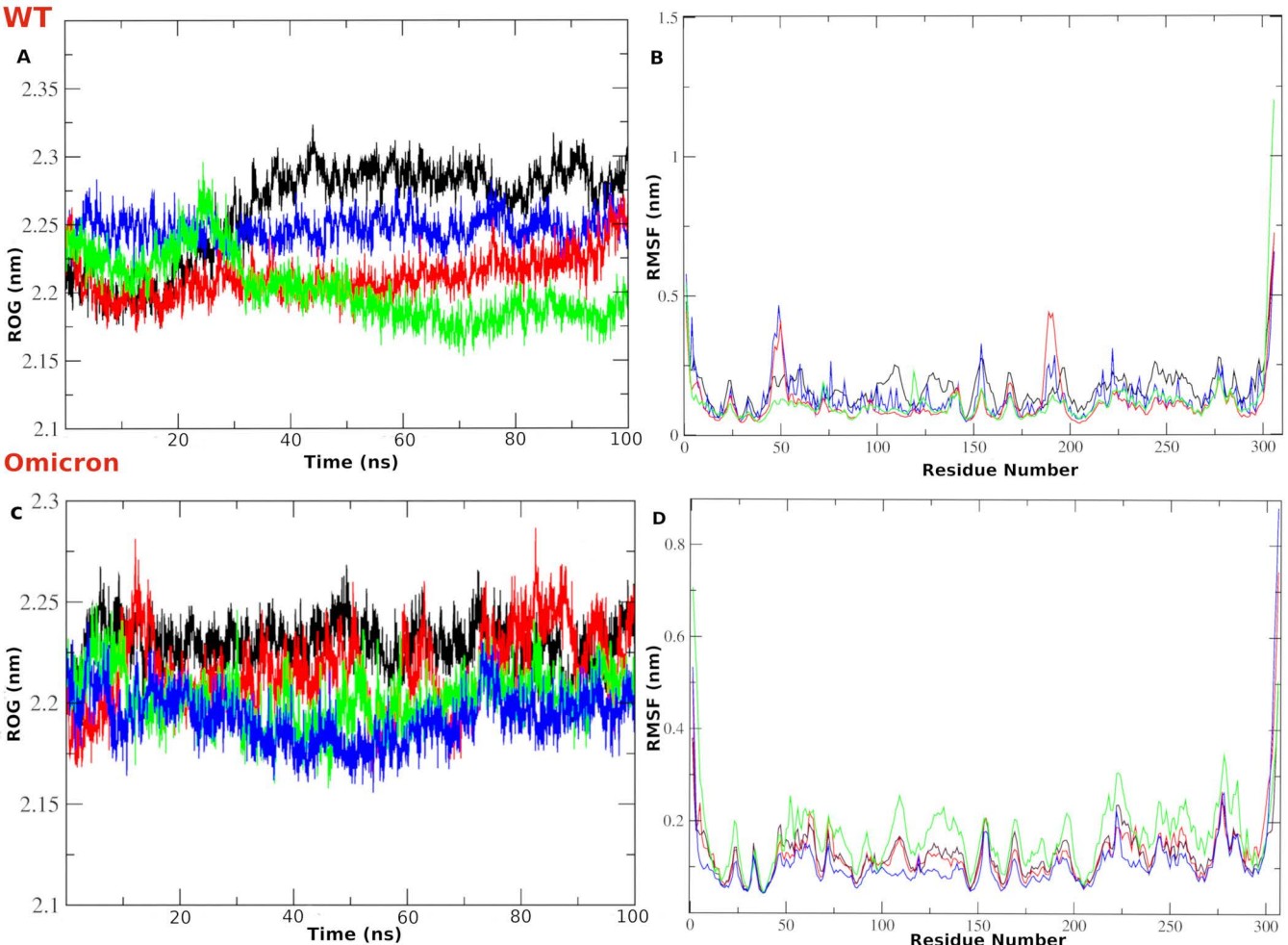

**Fig 11. The RoG and RMSF analysis of the MD simulation for the WTMpro complex systems.** (A) Protein radius of gyration (RoG). (B) Per-residue root mean square fluctuation (RMSF). For OMpro (C) Protein radius of gyration (RoG). (D) Per-residue root mean square fluctuation (RMSF). The color scheme is represented as: blue line is for the co-crystal ligand in both WT and Omicron. The red line is for compound **386**-WTMpro and compound **401**-OMpro complex systems. The green line is for compounds **329**-WTMpro and **541**-OMpro complex systems.

for the Mpro protein. The flexibility of these regions was greatly minimized in liganded systems. Interestingly, complex systems with compound **386** (red), co-crystal ligand (blue), and compound **329** (green) exhibited lower RMSF at the binding site amino acids compared to the unliganded system. This suggests a pronounced stabilization effect of the complexed ligands and their influence in reducing the conformational perturbations of the binding site residues.

## Binding free energy (ΔG) via MM GBSA calculation

Binding free energy (ΔG) for all the 4 complexes compound **386**, compound **329** with the WTMpro, and compound **401** and compound **541** with the OMpro were calculated for the whole 100 ns of the simulation and averaged. In addition, the two co-crystal ligands were calculated for comparison purposes. This would provide insights to estimate the overall binding strength. **Table 4** shows the binding free energy (ΔG) and its components for all complex systems. The free energies are calculated based on two main terms, the Gibbs free energy of the gas ($\Delta G_{GAS}$) and solvation free energy (Gsolv). The first term $\Delta G_{GAS}$ is composed of the sum

**Table 4. Binding free energy of ligand-Mpro complexes.**

| Complex WTMPro (7L14) OMPro (7TLL) | Energy (Kcal/mol) | | | | | | |
|---|---|---|---|---|---|---|---|
| | VDWAALS | $\Delta_{ELE}$ | $\Delta E_{GB}$ | $\Delta E_{SURF}$ | $\Delta G_{GAS}$ | $\Delta G_{SOLV}$ | $\Delta TOTAL$ |
| 7L14-XFD | −30.66 | −15.57 | 26.1 | −3.82 | −46.23 | 22.28 | −23.95 |
| 7L14-329 | −43.34 | 35.78 | −18.19 | −5.58 | −7.56 | −23.76 | −31.32 |
| 7L14-386 | −32.28 | 38.21 | −20.84 | −4.25 | 5.93 | −25.09 | −19.16 |
| 7TLL-nirmatrelvir | −48.79 | −37.88 | 54.77 | −5.98 | −86.67 | 48.79 | −37.88 |
| 7TLL-401 | −44.8 | −20.35 | 32.82 | −5.6 | −65.15 | 27.22 | −37.93 |
| 7TLL- 541 | −39.43 | −19.27 | 35.64 | −5.18 | −58.69 | 30.46 | −28.23 |

of electrostatic energy ($\Delta_{ELE}$) and van der Waals energy ($_{VDWAALS}$). The second term (Gsolv) is represented by the sum of generalized Born Polar solvation energy ($E_{GB}$) and the non-polar solvation energies $\Delta E_{SURF}$ [54]. For the WT complex systems, compound **329** complex showed superior binding compared to both the co-crystal ligand and compound **386**, with a ΔG value of −31.32 kcal/mol, compared to −21.15 kcal/mol and −19.16 kcal/mol for the co-crystal ligand and compound **386**, respectively. The major contributing component to this superior binding is the van der Waals energy ($_{VDWAALS}$) (**Table 4**). This highlights the importance of the hydrophobic groups and aromatic groups of the proposed binders. For the Omicron complex systems, both nirmatrelvir (co-crystal ligand) and compound **401** showed the best and comparable ΔG values of −37.88 kcal/mol and −37.93 kcal/mol, respectively. These results indicated that hit compounds had an acceptable range of ΔG and formed a stable complex with the WTMpro and its OMpro. Again, the van der Waals energy is the main contributing term for the superior binding to OMpro of compound **401**. To provide more insights, the total energy components for the best two binders were plotted as shown in Fig 12A and B for compounds **329**, and **401**, respectively. The energy composition analysis verified the strong interaction of compound **329** with Met49, Met165, Ser46, Asn142, Gly143, Thr45, and Cys145 of the WTMpro, as displayed in the individual energy contribution graph (**Fig 12C**). Also, the heatmap showed some residues to maintain the individual energy contribution throughout the 100 ns simulation's time, specifically, Met49 and Ser46 (**Fig 12E**). For OMpro, the individual energy contribution graph of compound **401** showed favorable interactions with Met165, Glu166, Met49, Asn142, His41, Thr25, and Ser144. Interestingly, the heatmap (**Fig 12F**) confirmed the maintenance of the energy contribution through the simulation's time.

## Minimum distance analysis

Elucidating more comprehension into the poses of compound **329**-WTMpro complex during the MD simulation, we calculated the minimum distance between the interacting ligand and protein atoms (**Fig 13**). Initially, the O atom of compound **329** showed H-bonding interactions with the HN atom of Gly143. This favorable interaction is mirrored in distances of ~0.3 to 0.4 nm (i.e., 3–4 Å), till near 60 ns and then the distance graph proposes that some dynamics affected the distance to reach ~1 nm. Interestingly, the position of the Met49 residue allowed the new interactions to appear from 40 ns between SD atom and two carbon atoms (C9 and C17) of compound **329**, reflecting the minimum distance starting from ~0.6 to 0.8 nm (i.e., 6–8 Å) (for C17) and then stabilized at 70 ns till the end. Likewise, ligand's O2 atom and NH atom of Ser46 displayed H-bonding interactions with comparable range of distances ~0.2 to 0.6 nm (i.e., 2–6 Å). Overall, this highlights the ability of compound **329** to exhibit mainly one pose during the simulation time until 60/70 ns, and another pose from 70 ns until the end. For the complex of compound **401** and OMpro (**Fig 14**), the ligand adapted

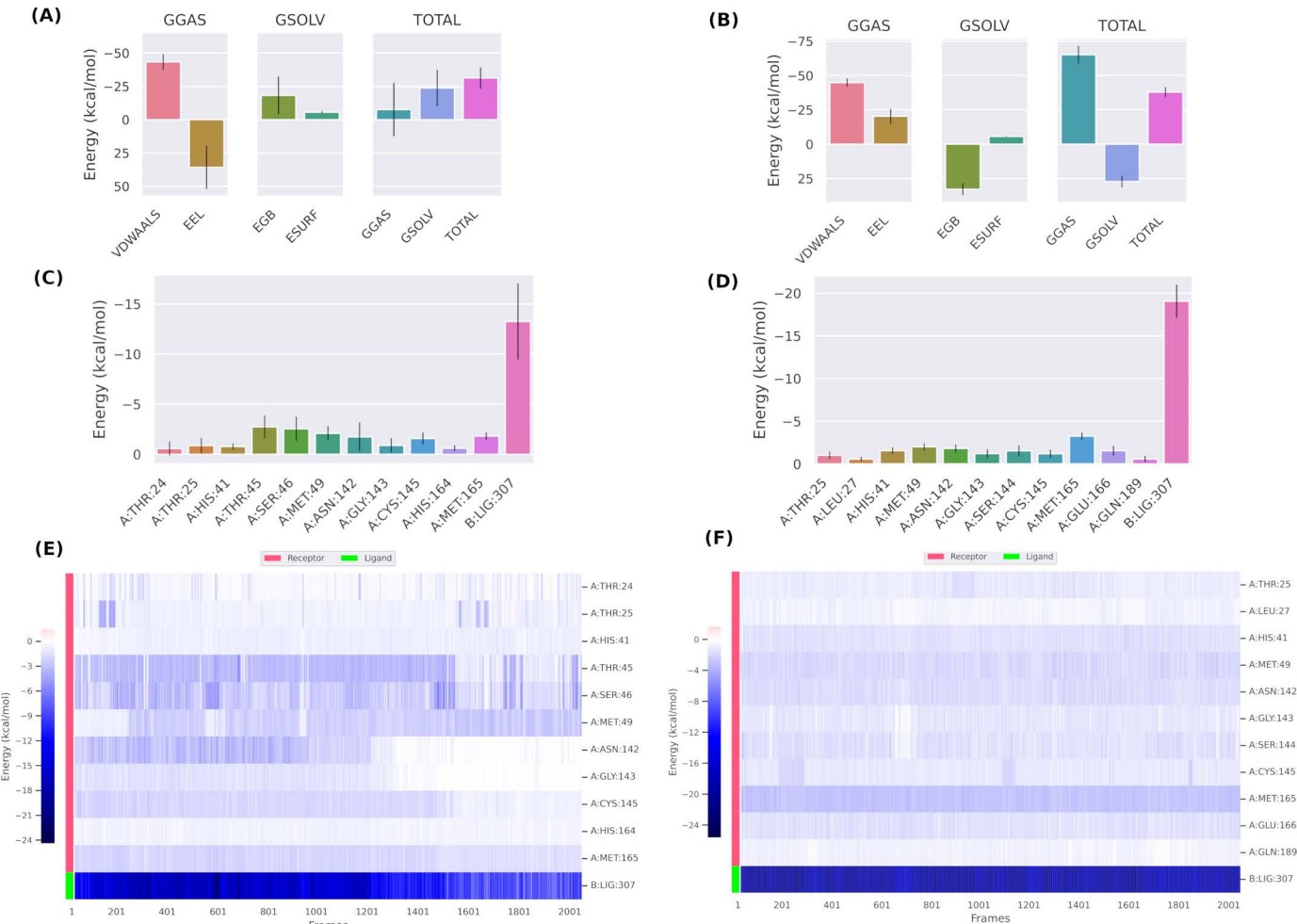

**Fig 12. Binding free energy ( ΔG) analysis using MM GBSA.** (A) Free energy components of compound **329** bound to WTMpro. (B) Energy components for compound **401** bound to OMpro. (C) Individual residue's energy contribution for compound **329**. (D) Individual residue's energy contributions for compound **401**. (E), (F). Heatmap showing the energy contribution of individual residues plotted against time for compounds **329**, and **401**, respectively.

one major pose along the simulation time. This is evident from the consistent distances (around ~ 0.2 nm) between the interacting atoms of the pyridone core (carbonyl oxygen atom) and the protein (H atom of Glu166). Likewise, stable distances indicated by the interacting atoms around ~ 0.2-0.3 nm were observed between SD of Met49 and H of Ser144 with C11 and N2 of compound **401**, respectively. Another stable distance between OD1 atom of Asn142 and C9 of compound **401** (around ~ 0.6 nm) indicates one major pose of the ligand.

## Conclusion

The aim of this study was a biphasic one, first to carry out an in-depth benchmarking investigation to recommend the best performing docking tool against SARS-CoV-2 Mpro on both wild type and the Omicron P132H variant. In order to do that, we assembled an active data set of small molecule inhibitors of the wild-type Mpro from the COVID-Moonshot database and literature, that was used to assess the performance of four docking tools, AutoDock Vina, FRED, PLANTS, and CDOCKER, we consequently created a top-notch DEKOIS 2.0 benchmark set for challenging these tools to discriminate between the actives and decoys.

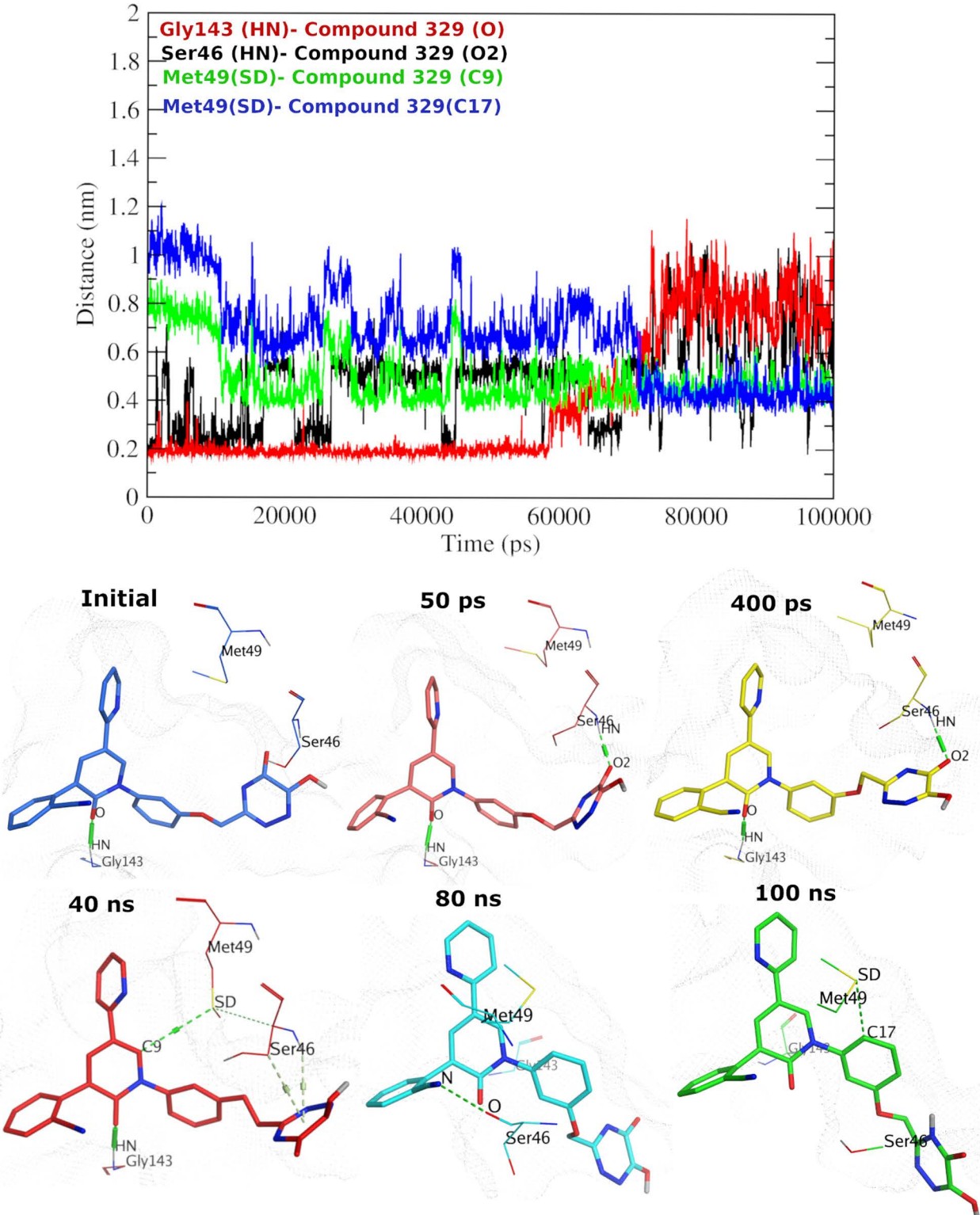

**Fig 13. The minimum distance graph of compound 329-WTMpro complex system.** Upper panel: The minimum distance graph throughout the MD simulation. Lower panel: snapshots at various simulation times.

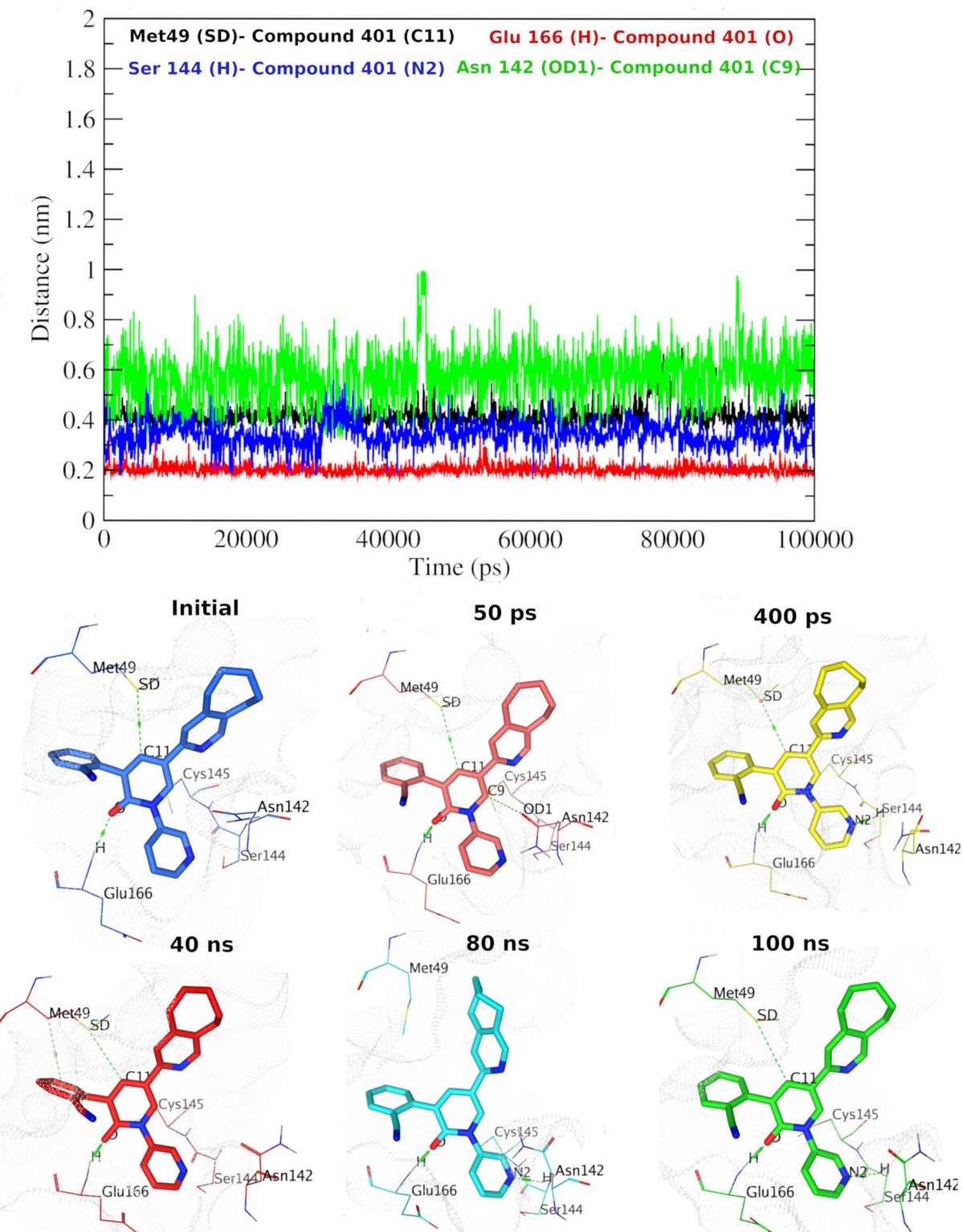

**Fig 14. The minimum distance graph of compound 401-OMpro complex system.** Upper panel: The minimum distance graph throughout the MD simulation. Lower panel: snapshots at various simulation times.

Benchmarking analysis assisted by pROC-Chemotype plots demonstrated that the most potent bioactives were highly enriched in the early docking ranks using AutoDock Vina for WTMPro and by both FRED and AutoDock Vina in the case of OMPro. The best enriched chemotype was detected to be cluster 1 (*N*-aryl pyridones), a well-known inhibitor belonging to which is Perampanel. In light of the benchmarking results, the SBVS was conducted for a targeted library transformed from Perampanel by scaffold hopping. The library was subjected to VS against both WTMpro and OMpro. Identifying two very well performing hits for each target, their stable binding was verified by MM GBSA calculations and molecular dynamics simulations. We thus identified compound **329** for WTMpro, and compound **401** for OMpro as very promising inhibitors and potentially good candidates for eliciting antiviral effect against SARS-CoV-2. Our work in progress is synthesizing those compounds for further extensive *in vitro* testing in the near future. In summary, this work provides an example of how to employ the power of benchmarking and SBVS to identify chemotypes for computer-assisted discovery of MPro inhibitors for the management of SARS-CoV-2.

## Supporting information

**S1 File. SI_file01 contains detailed description of the methodology of: Preparation of small molecules, preparation of the protein structures, and molecular dynamics simulations.** It also contains **Table A** for the active set chemotype clusters used for benchmarking, and **Table B** for the active set compounds used for benchmarking. It includes **Table C** showing the structures of **Table 1**. **SI_file02** contains the SMILES of the active and decoy molecules.
(ZIP)

## Acknowledgments

TMI acknowledges the cluster of Bibliotheca Alexandrina High-Performance Computing (BA-HPC) for awarding access to perform some calculations, and the lab of Prof. Frank Boeckler (Tuebingen University) for granting access to some computational tools. The authors acknowledge the support of OpenEye Scientific Software Inc. for offering a free academic license.

## Author contributions

**Conceptualization:** Noha Galal, Mohamed El-Hadidi, Reem K. Arafa, Tamer M. Ibrahim.

**Data curation:** Noha Galal.

**Formal analysis:** Reem K. Arafa, Tamer M. Ibrahim.

**Methodology:** Noha Galal, Botros Y. Beshay, Omar Soliman, Muhammad I. Ismail, Mohamed Abdelfadil, Tamer M. Ibrahim.

**Supervision:** Reem K. Arafa, Tamer M. Ibrahim.

**Visualization:** Noha Galal.

**Writing – original draft:** Noha Galal, Botros Y. Beshay, Omar Soliman, Muhammad I. Ismail, Mohamed Abdelfadil.

**Writing – review & editing:** Mohamed El-Hadidi, Reem K. Arafa, Tamer M. Ibrahim.

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
