## [Decision Letter · Decision Letter 0]

16 Dec 2024

PONE-D-24-54776Evaluating the Structure-Based Virtual Screening Performance of SARS-CoV-2 Main Protease: A Benchmarking Approach and a Multistage Screening Example Against the Wild-type and Omicron VariantsPLOS ONE

Dear Dr. Ibrahim,

Thank you for submitting your manuscript to PLOS ONE. After careful consideration, we feel that it has merit but does not fully meet PLOS ONE’s publication criteria as it currently stands. Therefore, we invite you to submit a revised version of the manuscript that addresses the points raised during the review process.

We look forward to receiving your revised manuscript.

Kind regards,

Ahmed A. Al-Karmalawy, PhD

Academic Editor

PLOS ONE

https://www.mdpi.com/1420-3049/28/3/1296

https://www.sciencedirect.com/science/article/pii/S0010482521002626?via%3Dihub

https://journals.plos.org/plosone/article?id=10.1371%2Fjournal.pone.0284301

In your revision ensure you cite all your sources (including your own works), and quote or rephrase any duplicated text outside the methods section. Further consideration is dependent on these concerns being addressed.

Reviewers' comments:

Reviewer's Responses to Questions

**Comments to the Author**

1. Is the manuscript technically sound, and do the data support the conclusions?

Reviewer #1: Partly

Reviewer #2: Yes

2. Has the statistical analysis been performed appropriately and rigorously? 

Reviewer #1: Yes

Reviewer #2: Yes

3. Have the authors made all data underlying the findings in their manuscript fully available?

Reviewer #1: Yes

Reviewer #2: Yes

4. Is the manuscript presented in an intelligible fashion and written in standard English?

Reviewer #1: Yes

Reviewer #2: Yes

5. Review Comments to the Author

Reviewer #1: The work is quite interesting and the findings are well put together. The only concern here is the time scale of the MD simulation. I was wondering if there were any replicates carried out to validate the MD simulations, mostly for sanity check? If not, then 100 ns of MD with CHARMM is inadequate sampling in my opinion. This is also reflected in certain structural analysis, where the system stability is seen towards the end of the simulations. Longer simulation times might provide greater insight into such situations.

Along with the analysis performed, looking into non-covalent interaction energies to look into important residues that take part in binding and are important for stability might be useful. Using energy decomposition analysis for such is a good benchmark.

Reviewer #2: The manuscript is well structured and relevant to its field and can be proceeded to publication.

Authors cold just consider provide descriptions for both the bond angles and distances of their depicted ligand-target interaction patterns. Specially for Hydrogen bonding, this type of compound-protein polar interaction should be presented within hydrogen bond distances as well as bond angles since hydrogen bond depend on both. Authors should mention the Hydrogen bond angles as well as their distances, since the strength of hydrogen bonding is based on both parameters in a way to ensure the adequacy of optimum hydrogen bonding.

6. PLOS authors have the option to publish the peer review history of their article (what does this mean? ). If published, this will include your full peer review and any attached files.

**Do you want your identity to be public for this peer review?** For information about this choice, including consent withdrawal, please see our Privacy Policy .

Reviewer #1: No

Reviewer #2: **Yes**

---

## [Author Response · Author response to Decision Letter 0]

24 Dec 2024

Response: We revised the PLOS ONE’s style requirements and addressed all of them.

Response: All findings are based on already published codes/methodologies. We cited them appropriately.

https://www.mdpi.com/1420-3049/28/3/1296

https://www.sciencedirect.com/science/article/pii/S0010482521002626?via%3Dihub

https://journals.plos.org/plosone/article?id=10.1371%2Fjournal.pone.0284301

In your revision ensure you cite all your sources (including your own works), and quote or rephrase any duplicated text outside the methods section. Further consideration is dependent on these concerns being addressed.

Response: We addressed them and cited the references mentioned.

Reviewers’ comments:

Reviewer #1: The work is quite interesting and the findings are well put together. The only concern here is the time scale of the MD simulation. I was wondering if there were any replicates carried out to validate the MD simulations, mostly for sanity check? If not, then 100 ns of MD with CHARMM is inadequate sampling in my opinion. This is also reflected in certain structural analysis, where the system stability is seen towards the end of the simulations. Longer simulation times might provide greater insight into such situations.

Along with the analysis performed, looking into non-covalent interaction energies to look into important residues that take part in binding and are important for stability might be useful. Using energy decomposition analysis for such is a good benchmark.

Response: We thank the reviewer for raising this point.

Indeed, the main focus of our study is to provide the community with an evaluation/assessment of the virtual screening performance via in-depth benchmarking analysis against the highly represented protein target in literature for COVID-19, the Mpro. The molecular dynamics protocol is an auxiliary approach and example of how to in-silico confirm the binding event of the proposed hits in the binding site of both WTMpro and OMpro.

Furthermore, Mpro is extensively studied in literature and employing MD simulation for a 100 ns is a standard performance in many studies especially in the context of assessing ligand-protein stability, e.g., PLOS ONE. 2023;18(4):e0284301. doi: 10.1371/journal.pone.0284301; and

https://www.tandfonline.com/doi/full/10.1080/07391102.2022.2148000

Nonetheless, we already performed a pilot MD simulation for 200 ns during our experimental setup for the co-crystal ligand (WT – PDB ID: 7L14), as shown in Figure R (please, see the "Response-to-reviewers" document). The RMSD fluctuations (blue line) did not differ greatly after 100 ns of simulation time.

Based on all the previous, we decided to set the MD simulation time for 100 ns, and therefore, we kindly appreciate the reviewer’s understanding in this matter. We also provided extensive analyses of the MD trajectories including, MMGBSA free energy calculations, per-residue energy contribution, minimum distance analysis and providing different screenshots at different time points.

Reviewer #2: The manuscript is well structured and relevant to its field and can be proceeded to publication.

Authors cold just consider provide descriptions for both the bond angles and distances of their depicted ligand-target interaction patterns. Specially for Hydrogen bonding, this type of compound-protein polar interaction should be presented within hydrogen bond distances as well as bond angles since hydrogen bond depend on both. Authors should mention the Hydrogen bond angles as well as their distances, since the strength of hydrogen bonding is based on both parameters in a way to ensure the adequacy of optimum hydrogen bonding.

Response: We thank the reviewer for his comment.

The H-bonding is depicted already in Figure 9 and 10 in the manuscript (H-bonding count). The native function of gromacs for H-bonding count considering both the distance and angle (please, see gromacs documentation). We quote the following from gromacs documentation: “A) Distance Criterion: The distance between the donor atom (typically a hydrogen bond donor such as an OH or NH group) and the acceptor atom must be less than or equal to rHB=0.35 nm. B) Angle Criterion: The angle formed by the hydrogen atom, the donor atom, and the acceptor atom must be less than or equal to αHB=30∘.”

Furthermore, based on the binding free energy (ΔG), in “Binding free energy (ΔG) via MM/GBSA calculation“ in the manuscript, the van der Waals energy was found to be the major contribution to the calculated ΔG for both WTMpro and OMpro complex systems. This minimizes the importance of providing more insights into some details for the H-bonding interactions.

Nonetheless, we already provided detailed distance analysis of the key H-bonding interactions for the WTMpro in the section “ Minimum Distance analysis”.

Thank you,

---

## [Decision Letter · Decision Letter 1]

21 Jan 2025

Evaluating the structure-based virtual screening performance of SARS-CoV-2 main protease: A benchmarking approach and a multistage screening example against the wild-type and Omicron variants

PONE-D-24-54776R1

Dear Dr. Ibrahim,

We’re pleased to inform you that your manuscript has been judged scientifically suitable for publication and will be formally accepted for publication once it meets all outstanding technical requirements.

Kind regards,

Ahmed A. Al-Karmalawy, PhD

Academic Editor

PLOS ONE

Reviewers' comments:

Reviewer's Responses to Questions

**Comments to the Author**

1. If the authors have adequately addressed your comments raised in a previous round of review and you feel that this manuscript is now acceptable for publication, you may indicate that here to bypass the “Comments to the Author” section, enter your conflict of interest statement in the “Confidential to Editor” section, and submit your "Accept" recommendation.

Reviewer #1: All comments have been addressed

Reviewer #2: (No Response)

2. Is the manuscript technically sound, and do the data support the conclusions?

Reviewer #1: Yes

Reviewer #2: (No Response)

3. Has the statistical analysis been performed appropriately and rigorously? 

Reviewer #1: Yes

Reviewer #2: (No Response)

4. Have the authors made all data underlying the findings in their manuscript fully available?

Reviewer #1: Yes

Reviewer #2: (No Response)

5. Is the manuscript presented in an intelligible fashion and written in standard English?

Reviewer #1: Yes

Reviewer #2: (No Response)

6. Review Comments to the Author

Reviewer #1: The reviewer's comments are addressed.

This paper is now suitable for publication and can be accepted.

Reviewer #2: (No Response)

7. PLOS authors have the option to publish the peer review history of their article (what does this mean? ). If published, this will include your full peer review and any attached files.

**Do you want your identity to be public for this peer review?** For information about this choice, including consent withdrawal, please see our Privacy Policy .

Reviewer #1: No

Reviewer #2: **Yes**

---

## [Editor Report · Acceptance letter]

PONE-D-24-54776R1

PLOS ONE

Dear Dr. Ibrahim,

I'm pleased to inform you that your manuscript has been deemed suitable for publication in PLOS ONE. Congratulations! Your manuscript is now being handed over to our production team.

Kind regards,

on behalf of

Associate Professor Ahmed A. Al-Karmalawy

Academic Editor

PLOS ONE